# A Longitudinal Study on Trajectories of Night Work and Sickness Absence among Hospital Employees

**DOI:** 10.3390/ijerph19138168

**Published:** 2022-07-03

**Authors:** Oxana Krutova, Aki Koskinen, Laura Peutere, Jenni Ervasti, Marianna Virtanen, Mikko Härmä, Annina Ropponen

**Affiliations:** 1Finnish Institute of Occupational Health, 00032 Helsinki, Finland; aki.koskinen@ttl.fi (A.K.); laura.peutere@uef.fi (L.P.); jenni.ervasti@ttl.fi (J.E.); mikko.harma@ttl.fi (M.H.); annina.ropponen@ttl.fi (A.R.); 2School of Educational Sciences and Psychology, University of Eastern Finland, 80101 Joensuu, Finland; marianna.virtanen@uef.fi; 3Division of Insurance Medicine, Department of Clinical Neuroscience, Karolinska Institutet, 171 77 Stockholm, Sweden

**Keywords:** night shift work, sickness absence, trajectory analysis, longitudinal, health care, employees

## Abstract

This study aimed to investigate trajectories of night shift work in irregular shift work across a 12-year follow-up among hospital employees with and without sickness absence (SA). The payroll-based register data of one hospital district in Finland included objective working hours and SA from 2008 to 2019. The number of night shifts per year was used in group-based trajectory modeling (GBTM). The results indicate that, among those who had any sickness absence episodes, the amount of night work decreased prior to the first SA. In general, trajectories of night shift work varied from stably high to low-but-increasing trajectories in terms of the number of shifts. However, a group with decreasing pattern of night work was identified only among those with sickness absence episodes but not among those without such episodes. To conclude, the identified trajectories of night work with or without sickness absences may indicate that, among those with sickness absence episodes, night work was reduced due to increasing health problems. Hence, the hospital employees working night shifts are likely a selected population because the employees who work at night are supposed to be healthier than those not opting for night work.

## 1. Introduction

The health care sector provides patient care around the clock. The average hours of work schedules are balanced every three weeks. However, due to varying patient flow, the order of the morning, day, evening, and night shifts is often irregular and varies between the weeks [1,2]. Furthermore, irregularity may be seen as consecutive work shifts, even without sufficient rest between shifts, and during scheduled time off [3]. Hence, this irregularity is manifested as varying shift start and end times, shift lengths, and rest periods between shifts [2,4,5,6]. For some employees, irregular working hours include night shifts [7,8,9,10]. In the Nordic countries, 10–14% of hospital employees worked frequently night shifts (>50 night shifts/year) in 2013 [2]. Previous studies have shown that night work increases the risk of acute and chronic health problems [1,11,12,13,14,15].

Sickness absence (SA) rates are generally high in the health care sector [16,17], and night work is also linked with sickness absence [18,19,20,21]. However, longitudinal assessments of night work and how the frequency of night work may change in anticipation of health-related outcomes such as SA are scarce. To date, most studies have focused on limited assessment time periods for night work [22,23,24] or have applied prospective study designs in which night work has been assessed at one time-point at baseline [19].

Besides the designs of earlier studies on night work, measures of SA have varied, limiting the comparability of findings [17,19,20,21]. For example, a retrospective longitudinal study in the UK indicated that frequent night work in a short time window (25–50% of the shifts in the past 7 days were night shifts) was associated with short SA (≤7 days), and more frequent night work (75%) increased the likelihood of long (≥7 days) SA [24]. In addition, several earlier studies have used one time-point night-work measure (an average for a year) and showed associations between working nights and very long (≥30 consecutive days) SA [19] or no association with long (>7 days) SA [21], whereas for fixed night work, both increased risk of short (≤8 days) SA as well as no association with long SA (≥9 days) have been shown [18]. However, these studies have not investigated the trajectories of night work across several years. Furthermore, the earlier studies were unable to identify the trajectories of night work overtime. The hypothesis of the current study is, in part, related to the healthy worker effect [25]; that is, those who work nights in irregular working hours might be a selected, healthier population which explains why increasing rates of SA might not be seen in night work over time. We hypothesised that, due to ageing, overall, the frequency of night work would decrease over time. Another hypothesis related to SA implied that the frequency of night work tends to decrease due to encountering SA [26].

Based on these specified hypotheses, we investigated the trajectories of night shifts across a 12-year follow-up among hospital employees in irregular shift work. We first focused on trajectories of night work in general in the whole material (sample 1) to see the general tendencies as regards night work. As the second aim, a history of night shift trajectories was identified for those with the first recorded SA episode during the follow-up and for those without SA. We focused separately on trajectories of night work for those employees with SA episodes (sample 2) and without them (sample 3) to see whether SA episode differently affects trajectories of night work. The association of age and sex were investigated in relation to the observed trajectories associated with both aims.

## 2. Materials and Methods

The data on working hours of one hospital district in Finland were obtained from the shift scheduling program Titania^®^, which is the payroll-based employer-owned digital programme for scheduling and monitoring working hours. The working hour data included start/end of work and absences including days off, sickness absences and other leaves, work unit, and type of shift work [6]. The data over the follow-up from 1 January 2008 to 1 November 2020 were used. In total, we acquired the data for 17,887 hospital employees working in the hospital district between 2008 and 2020. We excluded the year 2020 (*n* = 650 employees) due to incomplete working hour data for 2020. The sample formation, exposure time, and follow-up time for samples 1, 2, and 3 are presented in Figure 1.

In sample 1, all employees with ≥30 work shifts within a year in any year and those who had been at work at least in three consecutive years with ≥30 work shifts within each of those years during the exposure time were included (*n* = 5937 employees). Three consecutive years were included to assure exposure to the night work in the sample. Based on the annual number of night shifts, we identified night-shift trajectories over 12 years for sample 1.

In sample 2 (employees with SA), we retained the first criterion (≥30 work shifts within a year in any year). We identified the first SA episode of any length (*n* = 7919 employees with a sickness absence period). The mean length of the first SA episode was 4.6 days (standard deviation (SD), 12.1; range, 1–315). Then, the number of night shifts per year for the time from the first day of employment was calculated in sample 2 until the first SA episode. The 15 employees who did not have any working days from the first day of employment until the first SA episode (i.e., SA occurred already on the first day of employment) were excluded from the analysis. The final sample was 7904 employees with the first recorded SA episode. We focused only on the first SA episode to avoid reverse causation and to ease interpretation, as employees with repeated SA might change their working hours to cope with their health problems. Based on the annual number of night shifts, we identified trajectories for this group of employees until the first SA episode occurred over the follow-up from 2008 to 2019.

Sample 3 (employees without SA) included only those without SA and retained the first criterion (≥30 work shifts within a year in any year). Based on the annual number of night shifts, we identified trajectories for this group of employees that occurred over the follow-up period from 2008 to 2019 (*n* = 2021 employees).

The number of night shifts per year was used as a time-dependent covariate. Other covariates were sex and age.

### Statistical Analysis

We described samples 1, 2, and 3 with frequencies and means. Then, group-based trajectory modeling (GBTM) was performed. The GBTM allows for the identification of groups of individuals following similar progressions of night work over time and estimates the effects of covariates on trajectory shape, as well as on group membership. The GBTM enables subpopulations (clusters) existing within a studied population to be distinguished and described. We expected the trajectories of subpopulations to differ substantially from each other and from the average trajectory of the entire population. The flexmix package in R was utilised for analysis [27,28]. Finite mixture models are a popular method for modelling unobserved heterogeneity or for approximating general distribution functions [29,30].

To define the number of groups, we utilised the number of cases and the number of repeated measures (e.g., persons). The goodness of model fit was judged by running the procedure several times with several subpopulations. The Bayesian information criterion (BIC), Akaike information criterion (AIC), and average posterior probability (APP) were used as criteria to confirm the goodness of fit. Trajectory groups were identified for samples 1, 2, and 3. Trajectory analyses were conducted with R version 4.0.5 (The R Foundation for Statistical Computing, Vienna, Austria).

We used a multinomial logistic regression model for our analysis to estimate the association of covariates on trajectories for relative rate ratios (RRRs), with 95% confidence intervals (CIs). We compared associations for those with the first SA (sample 2) and those without SA (sample 3). This analysis was conducted with STATA 17.0.

## 3. Results

Most employees (83% in sample 1, 85% in sample 2, and 75% in sample 3) were women. On average, the employees were slightly older in sample 1 (39.8 years) than in sample 2 (37.5 years) or sample 3 (34.6 years). The employees had a mean of 12.1 night shifts per year (SD 23.1) in sample 1, 6.5 night shifts per year (SD 16.1) in sample 2, and 3.4 night shifts per year (SD 10.6) in sample 3.

Sample 1: Based on consecutive trajectory models with an increasing number of groups, the five-group trajectory solution was chosen based on model fit indices (AIC and BIC)(Table 1). The five identified groups can be seen in Figure 2.

Appendix A contains the mean frequency of night shifts from 2008 to 2019 in samples 1, 2, and 3. In sample 1 (all employees with at least 3 years at work), group 3 included 5% of employees and was characterised by a stably high number of night shifts during follow-up, on average, 73 night shifts per year (“stably high night shifts”). Group 2 included 17% of employees, who had a stably moderate number of night shifts, on average, 34 night shifts per year (“stably moderate number of night shifts”). Group 4 included 9% of employees with a moderate number of night shifts which were decreasing during 2008–2019 (“moderate and decreasing night shifts”). Group 5 included 9% of employees with a low but increasing number of night shifts (“low and increasing night shifts”). Group 1 included 60% of employees and was characterised by no night shifts (“stably no night shifts”).

For sample 2 (employees with SA), the chosen solution had three group trajectories based on model fit indices (AIC and BIC) (Table 2); the trajectories are shown in Figure 3.

In sample 2 (employees with SA), group 2 included 8% of employees with decreasing number of night shifts during 2008–2019 and, on average, 35 night shifts per year (“moderate and decreasing night shifts”). Group 3 included 24% of employees with decreasing number of night shifts during follow-up and, on average, very few (8) night shifts per year (“low and decreasing night shifts”). Group 1 included 68% of employees with no night shifts (“stably no night shifts”).

For sample 3 (employees without SA), the chosen solution had three group trajectories based on model fit indices (AIC and BIC) (Table 3); the trajectories are shown in Figure 4.

In sample 3 (employees without SA), group 2 included 7% of employees with a moderate and fluctuating number of night shifts during 2008–2019 and, on average, 30 night shifts per year (“moderate and fluctuating night shifts”). Group 3 included 17% of employees with a low number of night shifts during follow-up and, on average, very few (7) night shifts per year (“stably low night shifts”). Group 1 included 76% of employees with no night shifts (“stably no night shifts”).

Table 4 shows the association of age and sex with trajectory memberships in the three study samples. Older age was associated with a decreased likelihood of group membership in all trajectory groups in samples 1 and 2 in comparison to the group with “stably no night shifts”. However, men belonged more often to group 3 “stably high night shifts” in sample 1. Men also belonged more often to group 2 “moderate and decreasing night shifts” in sample 2. For other groups (except group 2 “moderate and fluctuating night shifts” in sample 3), male sex was associated with a lower likelihood of group membership (Table 4).

## 4. Discussion

This prospective study of one hospital district in Finland aimed to examine the trajectories of night shifts across a 12-year follow-up among 5937 hospital employees in irregular shift work. In addition, we separately studied the trajectories of night shifts among those with the first SA and among those without any SA. The trajectories indicated that, among all employees with more than three consecutive years at work, 60% did not work night shifts at all from 2008 to 2019. Of the studied employees, 5% work regularly around 70 night shifts/year, 17% had a stably moderate number of night shifts (about 35 night shifts/year), and 9% had a decreasing or low but increasing number of night shifts. An earlier study in the Nordic countries reported 10–14% work-frequent night shifts (>50 night shifts/year) in one year only—2013 [2]. In Finland, a study reported that 23% had night shifts, and 6.8% had 11–20 night shifts during the control time window of 30 days [10]. Compared with these studies, our study adds detailed numbers of night shifts across a rather long follow-up period of 12 years.

The hypothesis related to SA was that the frequency of night work decreases due to health issues. Register-based SA is assumed as a reliable indicator of health [26]. The trajectories limited to those with the first SA episode showed that still 68% had no night shifts, 8% had a high but decreasing number of night shifts, and 24% had a low but also decreasing number of night shifts before an episode of SA. Among employees who had no SA, the amount of night work was none for 76%, and 7% had a moderate but fluctuating number of night shifts, and 17% had a stably low number of night shifts. Thus, a group with decreasing pattern of night work was identified only among those with SA but not among those without SA. This may indicate that, among those with SA episodes, night work was reduced due to increasing health problems.

This is in line with earlier studies that have shown that night work increases the risk of SA [18,19,20,21]. Alternatively, our results may indicate a potential selection mechanism, i.e., those with health issues might not work nights, which is in line with the healthy worker effect hypothesis [25]. Furthermore, the differences in trajectories of night shifts across time and across samples in this study provided some support for this hypothesis. The healthy worker effect assumes that those who work nights in irregular working hours might be a selected population which may be seen in a relatively stable and large number of employees who did not work nights at all in the 12-year follow-up. This might also shed some light on why the increased risk of SA has not been seen in all earlier studies [22].

Another hypothesis was related to ageing, as previous studies suggest that older employees work fewer night shifts [23,31,32], i.e., the frequency of night work decreases over time. Some of the trajectories indicated a decreasing trend in the number of night shifts over time. However, older age was also associated with a lower likelihood of belonging to the trajectory groups with decreasing trend in night shifts.

Men were less likely to belong to trajectories with a stable number of night shifts, decreasing number of night shifts, or low and decreasing/increasing night shifts. Instead, men had a higher likelihood of belonging to trajectories with stably high or high but decreasing night shifts. This may indicate another selection mechanism. Men may either have more duties that require night work, or they volunteer to work nights. Both for workplaces and (occupational) health care, these results provide an understanding that those working night shifts should be given particular consideration in terms of recovery, lifestyle counselling, and occupational safety to maintain their workability, health, and well-being.

The strength of this study is in the usage of large, objective longitudinal data on working hours of one hospital district in Finland over the follow-up period from 2008 to 2019. Such a study with a detailed frequency of night shifts and assessment of SA might be among the first with such a comprehensive period of 12 years. Earlier studies have focused on limited assessment time periods for night work [22,23,24] or have applied prospective study designs in which night work has been assessed at one time-point at baseline [19]. Another strength of our study is in separate trajectories of night shifts for those with and without SA. Earlier studies have used one time-point night-work measure (for a year) and showed associations between working nights and SA [18,19,21]. However, these studies have not investigated the trajectories of night work across several years and were inconclusive and warrant the determination of how night work occurs with irregular working hours over time and whether the frequency of night work depends on encountering SA.

There are also some limitations of this study. Besides the designs of earlier studies on night work, measures of SA have varied, limiting the comparability of findings [19,22,23,24]. While earlier studies used various measures of SA, e.g., short (≤7 days), long SA (≥7 days) [21], or very long (≥30 consecutive days) SA [19], in this study, only the first SA episode without assessment of the length was evaluated. However, the trajectories of night work preceding the first SA might differ by the length of SA which should be taken into account in further studies. Although our sample was based on one hospital district in Finland, these results might be applicable to Finnish hospitals, in general, and to other Nordic countries with similar working hour arrangements in the hospitals and welfare system. However, generalizability might be less applicable to other countries.

## 5. Conclusions

This 12-year follow-up study suggests that a relatively large proportion of hospital employees do not work night shifts at all. Among those with sickness absence, the amount of night work decreased before sickness absence. Hence, hospital employees working night shifts are likely a selected population for whom support for workability, health, and well-being should be provided.

## Figures and Tables

**Figure 1 ijerph-19-08168-f001:**
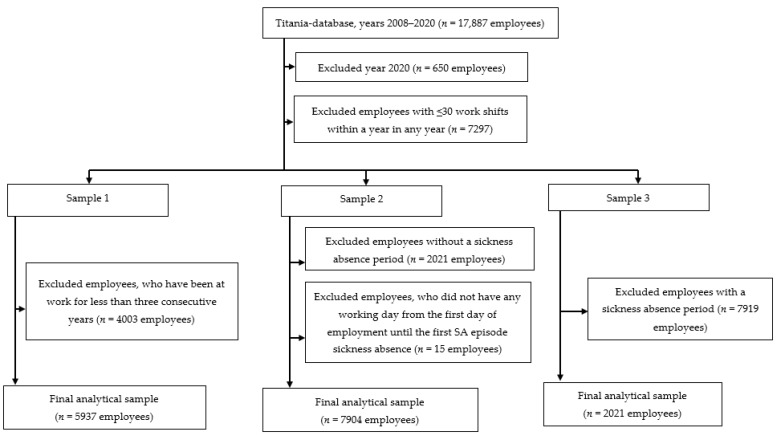
The flowchart of data selection for samples 1, 2, and 3.

**Figure 2 ijerph-19-08168-f002:**
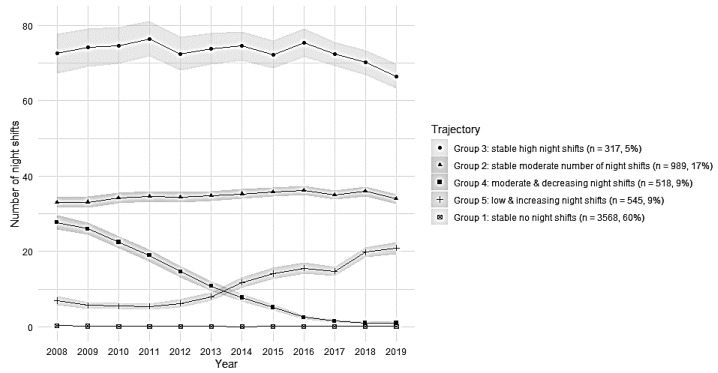
Trajectories of the mean annual number of night shifts with 95% CI in sample 1 (*n* = 5937 employees).

**Figure 3 ijerph-19-08168-f003:**
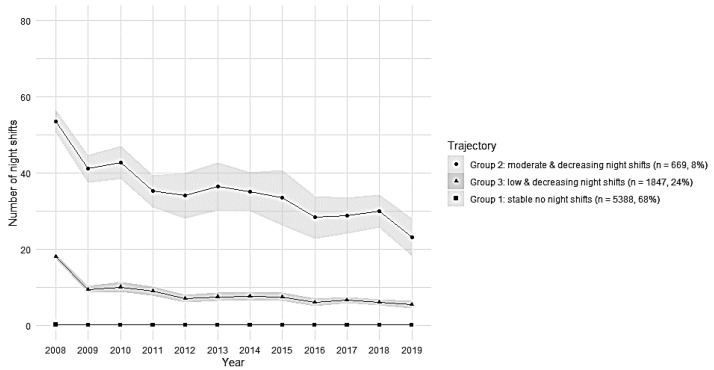
Trajectories of mean annual number of night shifts with 95% CI in sample 2 (*n* = 7904 employees).

**Figure 4 ijerph-19-08168-f004:**
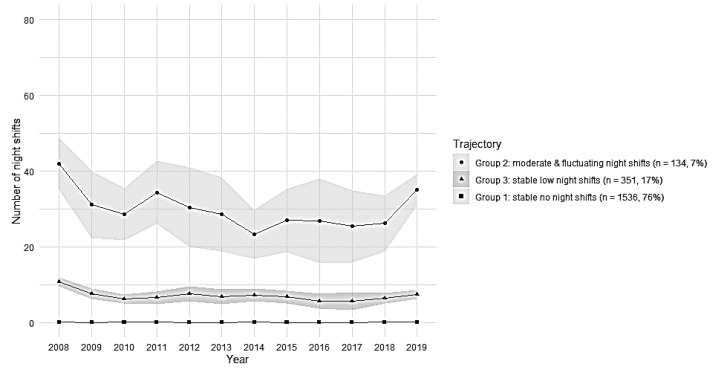
Trajectories of mean annual number of night shifts with 95% CI in sample 3 (*n* = 2021 employees).

**Table 1 ijerph-19-08168-t001:** The goodness of fit of group-based trajectory analysis for sample 1.

Model	Smallest Group Size	LL	AIC ^1^	BIC ^2^	APP ^3^
1 group	-	−841,647.5	1,683,301.0	1,683,327.4	-
2 group	33%	−281,503.0	563,019.9	563,081.5	0.99
3 group	16.3%	−211,076.7	422,175.5	422,272.3	0.97
4 group	6.7%	−180,798.0	361,626.0	361,757.9	0.95
5 group	5.3%	−167,268.0	334,573.9	334,741.1	0.95
6 group	5.3%	−167,267.9	334,573.7	334,740.9	0.87
7 group	16.3%	−180,798.0	361,626.0	361,757.9	0.97
8 group	5.5%	−211,076.8	422,175.6	422,272.3	0.88

^1^ Akaike information criteria; ^2^ Bayesian information criteria; ^3^ average posterior probability.

**Table 2 ijerph-19-08168-t002:** The goodness of fit of group-based trajectory analysis sample 2.

Model	Smallest Group Size	LL	AIC ^1^	BIC ^2^	APP ^3^
1 group	-	−159,558.9	319,123.8	319,146.5	-
2 group	23.9%	−55,261.7	110,537.5	110,590.5	0.94
3 group	8.4%	−40,774.9	81,571.9	81,655.2	0.79
4 group	14.4%	−45,123.9	90,269.9	90,353.1	0.77
5 group	14.4%	−45,123.9	90,269.9	90,353.2	0.77
6 group	6.4%	−37,400.7	74,839.4	74,983.2	0.51

^1^ Akaike information criteria; ^2^ Bayesian information criteria; ^3^ average posterior probability.

**Table 3 ijerph-19-08168-t003:** The goodness of fit of group-based trajectory analysis sample 3.

Model	Smallest Group Size	LL	AIC ^1^	BIC ^2^	APP ^3^
1 group	-	−29,494.218	58,994.44	59,013.36	-
2 group	19.3%	−9190.314	18,394.63	18,438.77	0.95
3 group	6.6%	−6800.363	13,622.73	13,692.10	0.74
4 group	6.5%	−6800.363	13,622.73	13,692.10	0.74
5 group	6.5%	−6800.363	13,622.73	13,692.10	0.74
6 group	6.5%	−6800.364	13,622.73	13,692.10	0.74

^1^ Akaike information criteria; ^2^ Bayesian information criteria; ^3^ average posterior probability.

**Table 4 ijerph-19-08168-t004:** Age and sex as predictors for night work trajectories.

Predictors	Sample 1 (*n* = 5937 Employees) ^1^							
	Group 3: Stably High Night Shifts		Group 2: Stably Moderate Number of Night Shifts		Group 4: Moderate and Decreasing Night Shifts		Group 5: Low and Increasing Night Shifts	
	RRR	95% CI	RRR	95% CI	RRR	95% CI	RRR	95% CI
Age (years)	0.96	0.95–0.97	0.94	0.93–0.94	0.97	0.96–0.97	0.93	0.92–0.94
Men (ref. women)	1.44	1.10–1.89	0.67	0.54–0.82	0.55	0.41–0.73	0.67	0.52–0.87
	**Sample 2 (*n* = 7904 employees) ^2^**							
					**Group 2: Moderate and Decreasing Night Shifts**		**Group 3: Low and Decreasing Night Shifts**	
Age (years)					0.99	0.98–0.99	0.96	0.96–0.97
Men (ref. women)					1.33	1.08–1.64	0.78	0.67–0.91
	**Sample 3 (*n* = 2021 employees) ^3^**							
					**Group 2: Moderate and Fluctuating Night Shifts**		**Group 3: Stably Low Night Shifts**	
Age (years)					0.99	0.98–1.01	0.97	0.96–0.98
Men (ref. women)					0.77	0.50–1.18	0.53	0.39–0.72

^1^ Group 1 (stably no night shifts) was used as reference. ^2^ Group 1 (stably no night shifts) was used as reference. ^3^ Group 1 (stably no night shifts) was used as reference.

## Data Availability

Data sharing not applicable.

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
