# Peer review of "A Longitudinal Study on Trajectories of Night Work and Sickness Absence among Hospital Employees"

_ijerph, 2022, doi:10.3390/ijerph19138168_

Round 1
Reviewer 1 Report
The document shows a structure according to what was requested, a coherence in the writing and theoretical and empirical contribution of novelty and relevance.
The document had been previously reviewed, and some suggestions were made, which were treated favorably.
This manuscript is a resubmission of an earlier submission. The following is a list of the peer review reports and author responses from that submission.
Round 1
Reviewer 1 Report
The sources cited at the beginning of the work are almost entirely the same used in the discussion.
It is recommended to carry out an exhaustive search in databases to strengthen the theoretical-empirical references in quantity and quality.
Verify that all the cited documents are in the list of references, since some were omitted.
Author Response
Dear reviewers,
Thank you for the comments and the opportunity to submit a revised version of our Manuscript ID: ijerph-1734232 entitled "A longitudinal study on trajectories of night work and sickness absence among hospital employees". We very much appreciate the points made by the reviewers and have now revised the manuscript accordingly. Our point-by-point responses to the specific comments made by the reviewers are below. The changes made to the text are highlighted in the manuscript.
Comments to the Author
The sources cited at the beginning of the work are almost entirely the same used in the discussion.
It is recommended to carry out an exhaustive search in databases to strengthen the theoretical-empirical references in quantity and quality.
Response:
Thank you for the suggestion. We have added seven theoretical-empirical references to the Introduction and Discussion sections. The added refences were:
- Angerer et al., 2017;
- Beltagy et al., 2018;
- Garde et al., 2020;
- Härmä et al., 2020;
- Karhula et al., 2020;
- Krane et al., 2014;
- Trinkoff et al., 2006.
Verify that all the cited documents are in the list of references, since some were omitted.
Response:
We’ve checked all the cited documents. We were not able to detect any missing referrals or that any would have been omitted. Some references have different names as “Chiara Dall’Ora, Ball, Redfern, & Griffiths, 2020”, “Chiara Dall’Ora et al., 2020” and “Dall’Ora, C., Ball, J., Redfern, O. C., & Griffiths, P. (2020). Night work for hospital nurses and sickness absence: a retrospective study using electronic rostering systems. Chronobiology International, 37(9-10), 1357-1364. doi:10.1080/07420528.2020.1806290”. This is the same reference, however.
Author Response
Dear reviewers,
Thank you for the comments and the opportunity to submit a revised version of our Manuscript ID: ijerph-1734232 entitled "A longitudinal study on trajectories of night work and sickness absence among hospital employees". We very much appreciate the points made by the reviewers and have now revised the manuscript accordingly. Our point-by-point responses to the specific comments made by the reviewers are below. The changes made to the text are highlighted in the manuscript.
Comments to the Author
An excellent and well-written paper, in my opinion. I have only few remarks and questions.
Response: We thank the reviewer for the positive feedback and constructive comments.
I had to read the description of how the samples were formed (p. 5, L1-20) a couple of time before I understood it (I hope). I think that the section would be a little clearer if you address these points: In Sample 1, why were employees what not been at work for at least three consecutive years excluded? Why was three years set as the limit? Why is the criterion not applied to Sample 2 and 3? In Sample 2, why did you choose the period from first day of employment until the first SA episode for analysis? Why not, for example, choose the period from the first SA episode until the second episode?
Response:
The reviewer has pointed a very relevant issue, sample selection. The sample 1 included the selection criteria of being at work at least three consecutive years to assure that there would be enough exposure time for night work since turnover at health care, but also absences both due to sickness and other reasons than sickness, such as maternity leave or education are rather frequent. What comes to the sample 2 and 3, we were specifically interested on first incident sickness absence. That is because sickness absences are often recurrent and there is a large chance that there would be another sickness absence spell after the first one. Recurrent sickness absences might change working hours and therefore focusing on the first sickness absence would represent a period without such effects. That also explains why we were not interested on the period between the first SA episode until the second episode. To make this clear in the manuscript, we added on page 5, paragraph 1:
“In sample 1, all employees with ≥30 work shifts within a year in any year and those who had been at work at least in three consecutive years with ≥30 work shifts within each of those years during the exposure time were included into the sample 1 (n=5,937 employees). Three consecutive years were included to assure exposure to the night work in the sample.”
Furthermore, we added to the page 5, paragraph 2: “We focused only on the first SA episode to avoid reverse causation and to ease interpretation, as employees with repeated SA might change their working hours to cope with their health problems.”
Results: As I understand it, Sample 2 and 3 are mutually exclusive, but Sample 2 and 3 together more or less constitute Sample 1. Therefore, I am surprised that employees with a stable high level of night shifts in Sample 1 (group 3) have disappeared from Sample 2 (Figure 2) and Sample 3 (Figure 3). It is also obvious in the supplemental Table 1A: Sample 1 group 3 has a mean level of annual night shifts around 75, while the highest mean levels in the other samples is around 40. How can it be?
Response:
We appreciate this notion which is correct. Sample 1 includes employees with ≥30 work shifts within a year and at work at least in three consecutive years. However, the Samples 2 and 3, the only criterion that were retained is the so called first criterion (i.e., ≥30 work shifts within a year in any year). Hence this has relaxed the second criterion in Sample 1 and explains why the characteristics and even night work may vary between our samples.
Discussion (L138-139): “Alternatively, our results may indicate potential selection mechanism, i.e., those with health issues might not work nights.” This sentence is confusing. I thought this was the main hypothesis.
Response:
We agree with the reviewer. This sentence is a part of section where we have discussed the hypotheses and findings of this study. To clarify this, we have revised the text on page 10, paragraph 3: “Alternatively, our results may indicate potential selection mechanism, i.e., those with health issues might not work nights, which is in line with the healthy worker effect hypothesis (Costa, 2010). Furthermore, the differences in trajectories of night shifts across time and across samples in this study provided some support for this hypothesis. The healthy worker effect assumes that those who work nights in irregular working hours might be a selected population which might be seen in relatively stable and large number of employees who do not work nights at all in 12-year follow-up.”
The authors use expressions like “…encountering sickness absence decreases the amount of night work…” (for example, in the abstract and in the conclusions). I find it problematic because it suggests that the results supports a certain causal direction of the association SA-night shifts. However, it is a cross-sectional study since two groups a simply compared (trajectories in Sample 2 versus those in Sample 3). In addition, in Sample 2 (employees with SA) the periods prior to SA, not following SA, are analyzed.
Response:
We thank the reviewer for this suggestion. However, in the trajectory analyses of the sample 2, that is the group with first sickness absence episode, the night work was followed until the SA episode. This is stated in the page 5, lines 11-13. Hence, our analysis was longitudinal for the assessment of night work i.e., night work was assessed over time until first SA episode. For sample 3 without SA, the follow-up of night work was longitudinal constituting the full follow-up from 2008 until 2019. However, since we agree with the reviewer that no direction of causation should be suggested, we have revised the text accordingly and removed mentions “encountering”. E.g., in abstract, the sentence reads now: “The results indicate that among those who had any sickness absence episodes, the amount of night work decreased prior to the first SA.” In the discussion, the sentence reads now: “Among employees who had no SA, the amount of night work was none for 76%, and….” Furthermore, the sentence in the conclusions was revised accordingly.
Minor remarks: Figure 1: The last line in one of the boxes below Sample 2 has almost disappeared. Line 9: “..per a year..” “..per year” Table 4: Footnotes 1-3 are missing.
Response:
Thank you for these. All have been corrected as suggested.
Reviewer 3 Report
Thank you for choosing me to review this manuscript.
In my opinion, the manuscript is very interesting. Its purpose is to analyze the effects of night shifts among hospital employees. One of the hypotheses put forward by the authors seems to refer to the effects of night shifts on healthy employees and / or with an illness.
The manuscript entitled "A longitudinal study on the trajectories of night work and sickness absence among hospital employees" is well structured and analyzes nursing, respectively night shifts and their effects on employees.
There are 25 references, of which a number of 9 are over 10 years old, 3 references between 5-10 years old and 13 from the last 5 years.
The names of some authors of this article appear in the references several times:
- Aki Koskinen - twice
-Virtanen - one time
- Mikko Härmä - 7 times
-Annina Ropponen - 6 times
The manuscript involves an investigation using the Titania program to schedule and monitor working hours at the hospital. The authors also present the results of other previous studies on night work in the hospital.
The authors developed a group-based trajectory modeling (GBTM) that allows the description of the existing subpopulations within the studied population. 3 information criteria were used, namely BIC, AIC and APP. The multinomial logistic regression model was used to estimate possible associations, with 95% IC inclusion intervals. The study was conducted over a period of 12 years and involved a number of 5937 people.
A strong point of the study is the use of data over a long period of time. There is also a limitation of the study, also recognized by the authors, namely the evaluation of only the first episode of absence caused by the disease, not the duration. In the opinion of the authors, it seems that this model of appreciation could be applied in hospitals in Finland.
The conclusions are in line with those presented by the authors in the manuscript.
The data in the tables is easy to interpret.
We did not find any references to the ethical approval of the study.
Lines 27-29: ,, Due to varying patient flow, working hours are often irregular. Irregularity is manifested as varying shift start and end times, shift lengths, and rest periods between shifts”.
Is the work schedule not the standard number of hours? What do you mean when you say that due to the variable flow of patients the work schedule is often irregular?
Lines 36-39 : ,,The goodness of model fit was judged by running the procedure several times with several subpopulations. The Bayesian Information Criterion (BIC), Akaike information criterion (AIC) and average posterior probability (APP) were used as criteria to confirm the goodness of fit”.
What made you choose 3 criteria to compare the samples?
Lines 101-108: ,,Alternatively, our results may indicate potential selection mechanism, i.e., those with health issues might not work nights. The differences in trajectories of night shifts across time and across samples in this study lend some support for the hypothesis of the healthy worker effect in night work “
Were you able to identify the number of these people? Or at least estimate the percentage? How did this population group affect the statistical analysis performed?
Lines 142-147: ,, In this study, only first SA episode without assessment of the length was evaluated”.
Do you think that the results obtained can change a lot if you evaluate the duration, not just the first episode of the disease?
Lines 159-161: ,,Although our sample was based on one hospital district in Finland, these results might be applicable to Finnish hospitals in general and to other Nordic countries with similar working hour arrangements in the hospitals and welfare system. However, the generalizability might be less to other countries”.
What exactly do you mean? What are your arguments on this issue?
Conclusions
Lines 185-186 : ,,This 12-year follow-up study suggests that a relatively large proportion of hospital employees do not work night shifts at all”.
Can you specify if this conclusion is found in the tables or graphs in the paper?
Author Response
Dear reviewers,
Thank you for the comments and the opportunity to submit a revised version of our Manuscript ID: ijerph-1734232 entitled "A longitudinal study on trajectories of night work and sickness absence among hospital employees". We very much appreciate the points made by the reviewers and have now revised the manuscript accordingly. Our point-by-point responses to the specific comments made by the reviewers are below. The changes made to the text are highlighted in the manuscript.
Comments to the Author
Thank you for choosing me to review this manuscript. In my opinion, the manuscript is very interesting. Its purpose is to analyze the effects of night shifts among hospital employees. One of the hypotheses put forward by the authors seems to refer to the effects of night shifts on healthy employees and / or with an illness.
The manuscript entitled "A longitudinal study on the trajectories of night work and sickness absence among hospital employees" is well structured and analyzes nursing, respectively night shifts and their effects on employees.
Response: Thank you for the overall positive feedback on our manuscript.
There are 25 references, of which a number of 9 are over 10 years old, 3 references between 5-10 years old and 13 from the last 5 years.
The names of some authors of this article appear in the references several times:
- Aki Koskinen - twice
- Virtanen - one time
- Mikko Härmä - 7 times
- Annina Ropponen - 6 times
Response: Thank you for pointing these. Referral to the earlier works of the authors is necessary due to the group’s activity in research on night work and the need to correctly refer to the database used in the present study. In the revised manuscript, we have added seven new theoretical-empirical references to the Introduction and Discussion sections:
- Angerer et al., 2017;
- Beltagy et al., 2018;
- Garde et al., 2020;
- Härmä et al., 2020;
- Karhula et al., 2020;
- Krane et al., 2014;
- Trinkoff et al., 2006.
The manuscript involves an investigation using the Titania program to schedule and monitor working hours at the hospital. The authors also present the results of other previous studies on night work in the hospital.
The authors developed a group-based trajectory modeling (GBTM) that allows the description of the existing subpopulations within the studied population. 3 information criteria were used, namely BIC, AIC and APP. The multinomial logistic regression model was used to estimate possible associations, with 95% IC inclusion intervals. The study was conducted over a period of 12 years and involved a number of 5937 people.
A strong point of the study is the use of data over a long period of time. There is also a limitation of the study, also recognized by the authors, namely the evaluation of only the first episode of absence caused by the disease, not the duration. In the opinion of the authors, it seems that this model of appreciation could be applied in hospitals in Finland.
The conclusions are in line with those presented by the authors in the manuscript.
The data in the tables is easy to interpret.
Response: Thank you again for reviewing our manuscript.
We did not find any references to the ethical approval of the study.
Response:
This study was fully based on administrative register data that the hospital district had permitted the access. Research using such data does not need to undergo review by an ethics committee according to Finnish legislation (Medical Research Act).
Lines 27-29: “Due to varying patient flow, working hours are often irregular. Irregularity is manifested as varying shift start and end times, shift lengths, and rest periods between shifts”. Is the work schedule not the standard number of hours? What do you mean when you say that due to the variable flow of patients the work schedule is often irregular?
Response: In this study, the data was based on one hospital district in Finland which provides both specialized outpatient- and inpatient care including emergency care. Hence some operations and/or wards operate 24/7 without appointments. This results variable flow of patients. Furthermore, although work schedule may include standard number of hours, there might be occasions where over time or sudden changes to schedules are needed due to above mentioned patient flow. Since we appreciate this comment, we have revised the text in the introduction, and it reads now: “Healthcare sector provides patient care around the clock. The average hours of the work schedules are balanced every three weeks. However, due to varying patient flow, the order of the morning, day, evening and night shifts is often irregular and varies between the weeks (A. H. Garde et al., 2020; Anne Helene Garde et al., 2019). Furthermore, irregularity may be seen as consecutive work shifts, even without sufficient rest between shifts, and during scheduled time off (Trinkoff, Geiger-Brown, Brady, Lipscomb, & Muntaner, 2006). Hence this irregularity is manifested as varying shift start and end times, shift lengths, and rest periods between shifts ….”
Lines 36-39: “The goodness of model fit was judged by running the procedure several times with several subpopulations. The Bayesian Information Criterion (BIC), Akaike information criterion (AIC) and average posterior probability (APP) were used as criteria to confirm the goodness of fit”. What made you choose 3 criteria to compare the samples?
Response:
As we utilized the flexmix-package in R, we followed recommendations provided by Grun & Leisch (2007) and Leisch (2004). We therefore have chosen these three indicators to confirm the goodness of fit of trajectory models.
Lines 101-108: “Alternatively, our results may indicate potential selection mechanism, i.e., those with health issues might not work nights. The differences in trajectories of night shifts across time and across samples in this study lend some support for the hypothesis of the healthy worker effect in night work.” Were you able to identify the number of these people? Or at least estimate the percentage? How did this population group affect the statistical analysis performed?
Response: We would kindly like to point the reviewer to the Figures 2-4 where the number of employees within each trajectory group are presented. Especially the comparison of sample 2 (with first SA episode) and 3 (no SA) indicate that there are different trajectories of night work. Furthermore, among the sample 2 (with first episode of SA), there were 2 trajectories with decreasing trend of night work whereas no such trajectories in sample 3 (without SA). All the trajectory groups had enough statistical power for multinominal regression analyses.
Lines 142-147: “In this study, only first SA episode without assessment of the length was evaluated”. Do you think that the results obtained can change a lot if you evaluate the duration, not just the first episode of the disease?
Response:
We agree with the reviewer that assessment the length of SA episode might be of interest. However, since we restricted the sample based on the fact that they had the first SA episode and studied the amount of night work before that, one may assume that the length of SA episode might have less effect than if we would have followed the employees prospectively after the first SA episode. However, if this would hold true and what would be the effect of the length of the SA episode, should be evaluated in the further studies. We have addressed this in the discussion now (on page 10, lines 188-190): “Perhaps the trajectories of night work preceding the first SA might differ by the length of SA which should be accounted in further studies.”
Lines 159-161: “Although our sample was based on one hospital district in Finland, these results might be applicable to Finnish hospitals in general and to other Nordic countries with similar working hour arrangements in the hospitals and welfare system. However, the generalizability might be less to other countries”. What exactly do you mean? What are your arguments on this issue?
Response:
To the best of our knowledge, both working hour arrangements and welfare system are relatively similar in the Nordic countries (see e.g., Garde et al. 2019, Larsen et al. 2020, Ropponen et al. 2017) than they are in other countries such as USA (US Sick Leave In Global Context 2021; Gimenez-Nadal et al. 2022) or in Asian countries (Kogi et al. 1989; Guo et al. 2020).
References used here:
Garde, A. H., Harris, A., Vedaa, O., Bjorvatn, B., Hansen, J., Hansen, A. M., . . . Harma, M. I. (2019). Working hour characteristics and schedules among nurses in three Nordic countries - a comparative study using payroll data. BMC Nurs, 18(1), 12. doi:10.1186/s12912-019-0332-4
Gimenez-Nadal, J.I., Molina, J.A. & Velilla, J. Commuting time and sickness absence of US workers. Empirica (2022). https://doi.org/10.1007/s10663-022-09534-z
Guo, M., Tang, K., & Wang, Z. (2020). Commuting time and sickness absence in China: Rural/urban variations and Hukou impacts. The Economic and Labour Relations Review, 31(1), 76-95.
Kogi, K., Ong, C. N., & Cabantog, C. (1989). Some social aspects of shift work in Asian developing countries. International Journal of Industrial Ergonomics, 4(2), 151-159.
Larsen, A. D., Ropponen, A., Hansen, J., Hansen, A. M., Kolstad, H. A., Koskinen, A., . . . Garde, A. H. (2020). Working time characteristics and long-term sickness absence among Danish and Finnish nurses: A register-based study. Int J Nurs Stud, 112, 103639. doi:10.1016/j.ijnurstu.2020.103639
Ropponen A, Vanttola P, Koskinen A, Hakola T, Puttonen S, Härmä M. Effects of modifications to the health and social sector's collective agreement on the objective characteristics of working hours. Ind Health. 2017 Aug 8;55(4):354-361. doi: 10.2486/indhealth.2016-0166. Epub 2017 Apr 14. PMID: 28420807; PMCID: PMC5546844.
US Sick Leave In Global Context: US Eligibility Rules Widen Inequalities Despite Readily Available Solutions Jody Heymann, Aleta Sprague, Alison Earle, Michael McCormack, Willetta Waisath, and Amy Raub Health Affairs 2021 40:9, 1501-1509
Conclusions
Lines 185-186: “This 12-year follow-up study suggests that a relatively large proportion of hospital employees do not work night shifts at all”. Can you specify if this conclusion is found in the tables or graphs in the paper?
Response:
Thank you for this question. Yes, these results can be found in Figures 2-4 indicating the trajectory group with “no night work” as the largest in all. For example, in Sample 1, group 1 included 60% of employees and was characterized by no night shifts (“Stable no night shifts”, Figure 2). In Sample 2, group 1 included 68% of employees with no night shifts (“Stable no night shifts”, Figure 3). In Sample 3, group 1 included 76% of employees with no night shifts (“Stable no night shifts”, Figure 4).
Reviewer 4 Report
Dear author(s),
First of all, thank you for the opportunity to review this manuscript. Please, see below some comments/suggestions for your evaluation if they can contribute to improve your work.
General comments
· The abstract is well written and contains the main elements of the paper. The chosen keywords complement the title and abstract, contributing to the study being found by interested researchers.
· The introduction is well-structured and provides the necessary elements for the reader to understand the research gap and the aim of the authors. The authors show the importance of the theme in an adequate way, citing appropriate references.
· The method is well explained and is suitable for both the proposed objectives and the existing research gap.
· The results are consistent with the literature and the methodology.
· The conclusions are supported by the results.
Suggestions
· The article addresses a very important point in studies on work regarding the health of workers, especially those who are in the healthcare sector and perform night work. The study is longitudinal and analyzes 12-year data.
· Line 73 – Delete “.”
· Figure 1 needs to be fixed (maybe it's a formatting issue): the text of some boxes are cut off and the arrows are not connecting them properly
· I understand that the "Statistical analysis" section is a subsection of "Materials and Methods"; thus, I would suggest numbering the sections to make this clear.
· The numbering of figures is incorrect: there are two "Figure 1" in sequence. This propagates throughout the entire manuscript. Please check, after correcting this, if the text also needs to be corrected to match the numbering of the figures/tables.
· Please consider differentiating the legend of figures/tables using a different formatting than the one used for the text
· I recommend that figures are scaled to maximize the space used, as I believe the visualization of graphics and data can be improved, which is important for understanding and readability.
· Please consider proposing future avenues for research.
· Please consider screening the IJERPH to see if there are any studies that might be related to the topic.
Author Response
Dear reviewers,
Thank you for the comments and the opportunity to submit a revised version of our Manuscript ID: ijerph-1734232 entitled "A longitudinal study on trajectories of night work and sickness absence among hospital employees". We very much appreciate the points made by the reviewers and have now revised the manuscript accordingly. Our point-by-point responses to the specific comments made by the reviewers are below. The changes made to the text are highlighted in the manuscript.
Comments to the Author
Dear author(s),
First of all, thank you for the opportunity to review this manuscript. Please, see below some comments/suggestions for your evaluation if they can contribute to improve your work.
General comments
The abstract is well written and contains the main elements of the paper. The chosen keywords complement the title and abstract, contributing to the study being found by interested researchers.
The introduction is well-structured and provides the necessary elements for the reader to understand the research gap and the aim of the authors. The authors show the importance of the theme in an adequate way, citing appropriate references.
The method is well explained and is suitable for both the proposed objectives and the existing research gap.
The results are consistent with the literature and the methodology.
The conclusions are supported by the results.
Suggestions
The article addresses a very important point in studies on work regarding the health of workers, especially those who are in the healthcare sector and perform night work. The study is longitudinal and analyzes 12-year data.
Response: We thank the reviewer for very positive evaluation of our manuscript.
Line 37 – Delete “.”
Response: Done as suggested.
Figure 1 needs to be fixed (maybe it's a formatting issue): the text of some boxes are cut off and the arrows are not connecting them properly
Response: Corrected as suggested.
I understand that the "Statistical analysis" section is a subsection of "Materials and Methods"; thus, I would suggest numbering the sections to make this clear.
Response: We changed the heading for the "Statistical analysis" section being highlighted with Italics instead of boldface as other subtitles. We hope this would add to the clarity.
The numbering of figures is incorrect: there are two "Figure 1" in sequence. This propagates throughout the entire manuscript. Please check, after correcting this, if the text also needs to be corrected to match the numbering of the figures/tables.
Response: Corrected as suggested.
Please consider differentiating the legend of figures/tables using a different formatting than the one used for the text
Response: Unfortunately, the formatting is done by the Journal’s manuscript center, not by us authors.
I recommend that figures are scaled to maximize the space used, as I believe the visualization of graphics and data can be improved, which is important for understanding and readability.
Response: Yes, we tried to maximize the space used for the Figures. According to the requirements of the IJERPH Figures can be still modified by editors.
Please consider proposing future avenues for research.
Response: We would kindly like to point the reviewer to see few sentences that we already have for this in the manuscript. On page 10, lines 179-182, reads: “However, these studies have not investigated the trajectories of night work across years and were inconclusive and warrant to identify how night work occurs in irregular working hours over time and would the frequency of night work depend on encountering SA.” Furthermore, on the same page (10), lines 188-190 we have added now: “Perhaps the trajectories of night work preceding the first SA might differ by the length of SA which should be accounted in further studies.”
Please consider screening the IJERPH to see if there are any studies that might be related to the topic.
Response: Thank you for the suggestion. We indeed considered other articles on the similar topic published in the IJERPH. One of these articles has already been considered in our paper (e.g. Szkiela, M., Kusideł, E., Makowiec-Dąbrowska, T., & Kaleta, D. (2020). Night shift work—a risk factor for breast cancer. International Journal of Environmental Research and Public Health, 17(2), 659).